# Noncoding RNome as Enabling Biomarkers for Precision Health

**DOI:** 10.3390/ijms231810390

**Published:** 2022-09-08

**Authors:** Jit Kong Cheong, Dimple Rajgor, Yang Lv, Ka Yan Chung, Yew Chung Tang, He Cheng

**Affiliations:** 1Department of Biochemistry, Yong Loo Lin School of Medicine, National University of Singapore (NUS), Singapore 117597, Singapore; 2Precision Medicine Translational Research Programme, Yong Loo Lin School of Medicine, National University of Singapore (NUS), Singapore 117597, Singapore; 3NUS Centre for Cancer Research, Singapore 117599, Singapore; 4MiRXES Lab, Singapore 138667, Singapore

**Keywords:** biomarkers, ncRNA, miRNA, lncRNA, circRNA, exosomes, circulation, liquid biopsy, cancer, precision health

## Abstract

Noncoding RNAs (ncRNAs), in the form of structural, catalytic or regulatory RNAs, have emerged to be critical effectors of many biological processes. With the advent of new technologies, we have begun to appreciate how intracellular and circulatory ncRNAs elegantly choreograph the regulation of gene expression and protein function(s) in the cell. Armed with this knowledge, the clinical utility of ncRNAs as biomarkers has been recently tested in a wide range of human diseases. In this review, we examine how critical factors govern the success of interrogating ncRNA biomarker expression in liquid biopsies and tissues to enhance our current clinical management of human diseases, particularly in the context of cancer. We also discuss strategies to overcome key challenges that preclude ncRNAs from becoming standard-of-care clinical biomarkers, including sample pre-analytics standardization, data cross-validation with closer attention to discordant findings, as well as correlation with clinical outcomes. Although harnessing multi-modal information from disease-associated noncoding RNome (ncRNome) in biofluids or in tissues using artificial intelligence or machine learning is at the nascent stage, it will undoubtedly fuel the community adoption of precision population health.

## 1. Introduction

Identification of disease-specific molecular landscapes for proper patient stratification is pivotal to the success of precision medicine and precision health. Precision medicine is an approach to healthcare that takes into account the genetic makeup and characteristics of each patient. In contrast, precision health and precision population health have a broader scope—they not only encompass precision medicine but also approaches that occur outside the clinical setting, such as disease prevention, health promotion activities and providing the right health intervention to the right people at the right time.

Up until recent years, RNA was implicated by Francis Crick’s central dogma as the key messenger between DNA and protein [1]. With the advent of new technologies, it is now known that the human genome encodes a vast repertoire of noncoding RNAs (ncRNAs), which were thought to be meaningless “dark matter”. The current understanding of ncRNAs may look like an intertwined mess of molecules, but collectively they exhibit architecture and coordination. This allows for the elegantly choreographed regulation of DNA and protein expression/functions. While ncRNAs constitute more than 90 percent of the RNAs synthesized from the human genome, only a subset of them have been discovered and characterized over the past two decades [2].

The utility of ncRNAs as clinical biomarkers has been explored in a wide range of human diseases and conditions including cancer (reviewed in [2]), cardiovascular diseases (CVDs), diabetes or other metabolic diseases, immunological disorders, neurological diseases and infectious diseases (reviewed in [3]) including COVID-19 [4,5]. Notably, the promise of ncRNAs in altering the clinical trajectory of human diseases is best demonstrated in the context of human cancer, where ncRNA research is making significant contributions to advance our understanding of the complexity of cancer, as well as the many challenges in its management and therapy.

MicroRNAs (miRNAs) are among the first ncRNAs, found two decades ago, to be associated with the onset and progression of cancer [6]. This knowledge paved the way for a plethora of research studies that explored diverse ncRNAs as biomarkers (reviewed in [7,8]) not only in tumor tissues, but also in a wide variety of human biofluids (more prominently known as liquid biopsy) and subcellular components, such as the exosomes that contain molecular payloads including ncRNAs.

This review examines how critical factors, such as the stability of molecular analytes, govern the success of interrogating ncRNA biomarker expression in liquid biopsy and tissue. It also explores why the harnessing of information revealed by these tiny ncRNAs could enable better clinical management of human diseases, particularly in the context of cancer. The review also discusses the challenges and opportunities that lie ahead for ncRNAs to become standard-of-care clinical biomarkers.

## 2. ncRNAs Are Disease-Relevant Molecular Analytes

The recent discovery of ncRNAs, mainly via next-generation sequencing (NGS) platforms, have led to a paradigm shift in the way we think about the central dogma of molecular biology. NcRNAs, which exist either as structural, catalytic or regulatory RNAs, have been found to control many biological processes in the cell. They are broadly classified as short (<200 nucleotides, for example, microRNA [miRNA]) and long (>200 nucleotides) ncRNA (lncRNA) based on their size [9,10]. Of the numerous ncRNAs encoded in the human genome, miRNAs, tRNA-derived small RNAs (tsRNAs), piwi-interacting RNAs (piRNAs), long noncoding RNAs (lncRNAs) and circular RNAs (circRNAs) have been implicated in human diseases, particularly in cancer [11].

Among these ncRNAs, miRNAs remain the most ideal candidate for disease-relevant biomarker discovery and development due to the characteristics elaborated ahead. Unlike the other ncRNA groups that each possesses more than 10,000 members, the entire human miRNome only consists of 2654 mature miRNAs [12]. Of which, more than 800 mature miRNAs have been validated experimentally via multiple platforms and functional characterization studies [12]. MicroRNAs are actively secreted into circulation by cells, thus allowing their expression profiles to be easily extractable from liquid biopsy. They also show remarkable stability in biofluids, but exhibit dynamic changes during disease development [13]. Growing evidence suggests that alterations in miRNA expression is highly correlated with disease progression and burden, especially in the context of cancer [14,15]. LncRNAs, on the contrary, are usually present in ultra-low abundance in the circulation. This poses a daunting technical challenge to accurately quantify them using existing clinical laboratory tools, thus hindering their development as novel molecular biomarkers for disease diagnosis and prognosis (reviewed in [16,17]). The suboptimal specificity of lncRNAs is another major limiting factor that undermines their potential to become standard-of-care clinical diagnostic tests [18,19]. This is compounded by the lack of understanding of the many functions of all disease-associated lncRNAs and their gene target networks, thus hindering their fullest potential as diagnostic biomarkers. Similar issues are also occluding the clinical utility of circRNA biomarkers for disease diagnosis and prognosis.

Other than cancer, ncRNA biomarker discovery and development is gaining momentum in other human diseases. Expression profiling and the functional characterization of various ncRNAs, particularly miRNAs, have been recently explored in neurodegenerative disorders such as Alzheimer’s disease [20], spinal cord injury [21], epilepsy [22], rare and neglected diseases, such as leishmaniasis, African trypanosomiasis and leprosy (reviewed in [3]), as well as the diagnosis and/or prognosis of infectious diseases caused by viruses [23,24,25,26], including severe Acute respiratory syndrome–related coronavirus 2 (SARS-CoV-2) [4,5]. In this instance, it is envisaged that early detection of infectious diseases using ncRNAs related to viral-host response can not only facilitate timely triaging of patients for closer monitoring and therapy selection or prescription, but also help to curb the spread of disease by facilitating an isolation of cases.

More recently, ncRNA biomarkers have also shown to be valuable in the clinical management of a growing list of lifestyle diseases, including cardiovascular and metabolic disorders [27,28] (comprehensively reviewed in [29]). These ncRNA biomarkers, particularly the miRNAs, enable early detection of stroke [30], diabetic retinopathy [31] and type 2 diabetes mellitus [32]. They also facilitate cardiovascular disease (CVD) risk assessment of patients with rheumatoid arthritis [33]. Notably, circulating miRNAs have also been shown to combine synergistically with NT-proBNP to identify subtypes of heart failure with greater accuracy [34].

## 3. Liquid Biopsy as Surrogate for Tissue for Molecular Profiling

Detection of molecular biomarkers in dysfunctional organs or tissues has been the cornerstone of modern pathology to inform the extent of the disease. For instance, invasive surgical biopsy from tumor lesions has been routinely performed on cancer patients over the past few decades to rule out malignancy. Besides determining the type of cancerous cells, tissue-based molecular assays that are often performed individually also aid in cancer staging and grading. This costly and time-consuming approach is largely conducted by highly qualified lab technologists and reviewed by pathologists in clinical pathology labs. Furthermore, insufficient tissue biopsy samples from small tumors [35], false positivity associated with the preservation of tissue [36], tumor heterogeneity (reviewed in [37]) and variability in the results of different biopsies (reviewed in [38]) have also limited the overall performance of many tissue-based clinical assays. Although the issue of tumor heterogeneity could be circumvented by new multiplex technologies, such as spatial transcriptomics or other single-cell sequencing approaches, to determine the entire cellular landscape of the tumor microenvironment [39], the genetic profile of a tumor may change dynamically over time as a result of its natural evolution or response to therapy (reviewed in [40,41,42,43]). This necessitates repeated, longitudinal tumor biopsies to monitor disease progression or treatment response. However, this approach is unlikely to be feasible to most cancer patients, especially those with tumors that are either inaccessible or have metastasized to distant organs or tissues [44].

Due to these limitations (summarized in Table 1), detection of clinically relevant biomarkers in biofluids obtained by minimally invasive procedure (now widely known as liquid biopsy) has gathered significant interest for early disease diagnosis and surveillance over the past few years. This is exemplified by intense efforts to screen for circulating tumor cells (CTCs), CTC genomic DNA (gDNA) or tumor-derived products, such as circulating tumor DNA (ctDNA) or cell-free RNA (cfRNA) [45], in blood-based samples collected from cancer patients [46]. These analytes are usually detected in liquid biopsy from patients with advanced cancer [47]. However, early detection of disease-associated biomarkers is pivotal for the timely intervention of human diseases.

As demonstrated in the recent study conducted by Cohen and co-workers, ctDNAs were shown to combine synergistically with protein biomarkers to dramatically improve the diagnostic performance of the CancerSEEK multi-analyte blood test [47]. While ctDNAs provide high diagnostic specificity as they are released into circulation by the dying cancer cells in advanced-stage tumors, cancer protein biomarkers improve diagnostic sensitivity as they are actively secreted from cancer cells in early-stage tumors. Similarly, the global race to develop cost-effective clinical assays to detect ncRNAs in minimally invasive liquid biopsy has intensified over the past few years (reviewed in [48]).

ncRNAs, particularly miRNAs and circRNAs, exhibit great potential to fulfil many of the characteristics of a good biomarker, such as stability [49,50,51], availability in biofluids [52,53], readily detected by routinely used, cost-effective methods, such as RT-qPCR [54], even at the onset of human diseases. This is attributable to their active secretion from cells into the circulation in various forms, including encapsulation in extracellular vesicles/exosomes, protein- or lipid-bound. To date, this new generation of molecular analytes have been found in a growing list of biofluids, such as saliva, plasma, serum, blood, urine, sputum, cerebrospinal fluid, bile, gastric juice (reviewed in [55]).

## 4. Extracellular Vesicles/Exosomes: Valuable Biological Information Packages in Biofluids

Extracellular vesicles (EVs) are heterogeneous populations of nano- to micro-sized endosome-derived membrane vesicles that carry diverse molecular payloads, including nucleic acids (DNA, mRNA and ncRNAs), carbohydrates, lipids and proteins [56] (reviewed in [57,58]). They are actively secreted out of the cell in a dynamic manner to mediate cell-to-cell communication. Exosomes (30–200 nm in size) are the smallest subtype of EVs that are actively exported into circulation by living cells [59]. Their morphological diversity has been observed in various biofluids [60], indicating that subpopulations of exosomes carry different molecular effectors to control a multitude of biological functions across the human body.

Exosome enrichment from liquid biopsy has been shown to improve the signal-to-noise ratio of disease-relevant biomarkers [61]. Ultracentrifugation is the gold standard to isolate the nano-sized exosomes from liquid biopsy at present. However, this separation technique is time-consuming and it requires costly equipment and highly skilled lab personnel [62]. Furthermore, low sample throughput as well as poor exosome yield and quality of exosomes also limit the prospect of integrating ultracentrifugation into existing clinical lab testing workflow [63].

Besides ultracentrifugation, ultrafiltration and size-exclusion chromatography (SEC) are promising technologies that have been used to extract exosomes of various sizes and molecular weights. Although they have been shown to yield a high purity of exosomes within a shorter duration as compared to ultracentrifugation, these tiny exosomes are frequently lost during ultrafiltration or may be contaminated with lipoprotein during the SEC separation [64]. Lipoproteins mimic exosomes in size, thus leading to an overestimation of exosome abundance [65,66].

More recently, immunoaffinity-based approaches have also been developed to enrich exosomes from biospecimens. These methods typically use magnetic beads-conjugated specific antibodies, which are time-consuming and costly to produce, to target antigens commonly expressed on the surface of exosomes, including tetraspanins (CD9, CD63, CD81), lysosomal proteins (LAMP-2B), cell adhesion proteins (EpCAM, CD166), growth factor receptors (EGFR), integrins, multi-vesicular body (MVB) biogenesis-associated proteins (TSG101), etc. [67,68]. Similar to ultrafiltration and SEC, the immunoaffinity-based methods might not capture all exosomes that exist in the biospecimen because a subset of exosomes may not possess the canonical surface markers.

To increase the exosome capture efficiency of immunoaffinity-based methods, one could conjugate beads with bispecific or multiple antibodies that capture two or more exosome surface markers simultaneously. Alternatively, the fundamental principle of these various techniques can be adapted to newer platforms that are driven by microfluidics or nanotechnology to extract exosomes with high yield and purity. For instance, exosomes are separated in microfluidics systems based on their physical and chemical properties [69]. It has been demonstrated that the integration of acoustics into microfluidics devices can yield high-quality exosomes in a faster and more cost-effective manner [70].

Interestingly, polyethylene glycol (PEG)-based methods that are routinely used to isolate viruses have also been shown to provide an inexpensive and efficient alternative to purify exosomes from biospecimens. This is largely attributed to similar biophysical properties shared by exosomes and virus particles [71]. Various other exosome isolation technologies have also been developed (reviewed in [72]) and their advantages and disadvantages have been comprehensively reviewed [73,74,75,76,77,78].

At present, exosome isolation and characterization remain a nascent area that is poised for exponential growth in our ability to intercept exosomes in organ-specific biofluids for the detection of disease-associated biomarkers. Although exosomes are found in a wide variety of biofluids including blood (serum and plasma), urine, saliva, tears, semen, peritoneal lavage, bronchoalveolar lavage (BAL), etc. (reviewed in [79]), relatively little is known about the abundance of subpopulations of clinically-relevant exosomes in these biofluids and the dynamic composition of molecular payloads in the onset and progression of many medical conditions.

Notably, the miRNA profile of tumor cells has been shown to be highly similar to that of their secreted exosomes [80], suggesting that exosomal miRNAs can indeed provide clinically valuable information for the detection of human diseases. Several research groups have recently collated and curated the expression of EV-associated biomarkers in various biospecimens across many human medical conditions and organized the information in databases, such as EVmiRNA (miRNA-specific) [81], EVAtlas (ncRNA-specific) [82] and Vesiclepedia 2019 (RNA, proteins, lipids and metabolites) [83]. Such efforts not only enable the generation of new hypotheses to identify actionable biological pathways that promote the disease state, but also facilitate the development of miRNA/ncRNA-based in vitro diagnostics.

Although organ-specific biofluids in the proximity of the diseased cells or tissues can be a rich source of exosomes that carry clinically relevant biomarkers, one has to overcome biological and/or technical challenges associated with the extraction of exosomes from different biofluids. Urine, for instance, is typically a less ideal source of exosomal biomarkers (except for urological diseases) due to glomerular filtration. Furthermore, it is frequently collected in huge volumes that necessitates the use of secondary exosome concentration techniques to overcome the over-dilution of exosomal biomarker signals. To tackle these issues, Chen and co-workers developed an efficient exosome detection method via the ultrafast-isolation system (named EXODUS) that allows automated label-free purification of exosomes from biofluids, such as urine [84]. They demonstrated that exosomes could be purified from urine samples of 113 patients by negative pressure oscillation and double coupled harmonic oscillator-enabled membrane vibration. They further identified the genetic sources of urinary exosomes and showed that urinary exosomes are intensively involved in immune activities in cancer development [85].

Several key characteristics of exosomes make them ideal vehicles for biomarkers for clinical applications (Table 1). These include: (1) they are actively secreted by all cells (reviewed in [86]), although little is known about the difference in their shedding rate across all cell types in the human body, (2) they are biologically stable, (3) they shield their molecular payloads (DNA, RNA, proteins, lipids and carbohydrates) from degradative enzymes [87], thus allowing for multi-analyte analysis to increase the sensitivity and specificity of the clinical assay [88] (reviewed in [89]) and (4) their involvement in disease onset and/or progression (reviewed in [90]). For instance, exosomes have been implicated in different stages of cancer and its response to drug treatment, including growth of the tumor, suppression of immune response, induction of angiogenesis, metastasis and resistance to therapy [91,92] (reviewed in [93]).

It has been found that cancer cells produce more exosomes than non-cancerous cells [94,95]. The size and morphology of exosomes also vary between cancer patients and healthy controls [96]. Additionally, molecular analytes encapsulated in exosomes, such as exosomal DNA, RNA and/or ncRNA (exoDNA, exoRNA, exo-ncRNA) have been shown to synergize with each other or with free-floating circulating tumor DNA (ctDNA) or oncoproteins such as carcinoembryonic antigen (CEA) to improve cancer diagnosis, thus further enhancing the discriminatory power of exosomes [97,98,99,100,101,102,103,104]. Like ctDNA, these exosomal secondary messengers may also carry cancer-specific modifications that can aid in the detection of minimal residual disease (MRD) (reviewed in [57,105,106]).

## 5. Challenges and Opportunities for Clinical Applications with Exosomal ncRNA

Despite their huge potential in the patient care continuum and a projected market value of more than USD 50 billion in 2026 and beyond, data derived from exosomes and their molecular analytes [61] remain rather inconsistent [107]. This is likely due to the inconsistency in techniques used for sample pre-analytics (e.g., input sample collection and processing), exosome isolation, purification and quantification [108]. Thus far, the past literature has mainly focused on technical aspects of exosome capture as well as the function characterization of exosomes.

To advance exosomes and their molecular analytes from bench to bedside, it is necessary to establish international guidelines to govern the isolation or even synthesis of EV/exosome for various clinical applications. For instance, the International Society of Extracellular Vesicles (ISEV) published and also recently updated the minimal information for studies of EVs (MISEV) to standardize EV nomenclature, sample collection and pre-processing, EV separation and concentration, characterization, functional studies and reporting requirements/exceptions [109,110]. Such standards will lay a strong foundation for us to harness the full potential of exosomes to improve the clinical outcomes of patients.

With the aid of new technologies, it is now possible to analyze EVs (including exosomes) at the single particle level to better understand their biogenesis, correlate markers for higher specificity and connect EV cargo with the source or destination (comprehensively reviewed in [111]). Analysis of EVs/exosomes at the single particle level may hold the key to the establishment of disease-relevant diagnostic biomarkers, as the existing bulk of EV/exosome-based approaches falls short in addressing the specificity issue for biomarker identification. It remains to be seen whether single EV/exosome analysis will become a fundamental molecular technique that is widely adopted by the scientific community, much like single-cell RNA sequencing (scRNA-Seq) a decade ago [112].

## 6. Harnessing ncRNAs to Enhance Disease Management

Recent technological advancements have significantly enhanced the understanding of how ncRNAs modulate gene expression. While high-throughput NGS and microarray enable the detection of novel genetic alterations and gene expression changes in biospecimens, PCR-based technologies provide real-time, quantitative, sensitive and more robust ncRNA expression profiling to identify ncRNA biomarkers for the enhancement of current disease management. This includes the use of ncRNA biomarkers to complement existing standard-of-care diagnostics for early detection or screening of disease, disease subtyping, disease prognostication, treatment response prediction, treatment selection, as well as evaluation of residual disease (graphical abstract). As elaborated below with examples, how ncRNAs can serve as useful biomarkers for dynamic disease monitoring in the cancer care continuum (Figure 1) is discussed.

### 6.1. Early Detection/Screening

Early detection with timely interventions has been shown to effectively reduce disease mortality [152]. Detection of ncRNAs in liquid biopsy has been explored in a wide range of cancers, including lung cancer [14,15,113,114,115,116,117], breast cancer [118,119,120], colorectal cancer (or CRC) [121,122], gastric cancer [123] and pancreatic ductal adenocarcinoma [124]. Liquid biopsy facilitates early detection of diseases through increased uptake of population screening, where clinically relevant ncRNAs, such as miRNAs, can be readily detected in minute volume of biofluids decades prior to the development of disease symptoms [153].

### 6.2. Tumor Subtyping

ncRNAs also aid cancer subtype stratification, which is critical for treatment prescription. For example, miRNA classifiers that could differentiate subtypes of renal cell carcinoma using tissue samples have been identified [125]. Similarly, miRNA signatures have been found for different lung cancer subtypes (using preoperative cytologic samples) [126], papillary versus follicular thyroid cancer (using plasma exosomes) [127], local versus metastatic breast cancer (using plasma) [128] and early versus advanced stages of breast cancer (using data from the cancer genome atlas (TCGA) database) [129]. LncRNA and circRNA classifiers have also been developed for tumor differentiations in gastric cancer (using liquid biopsies) [130] and to identify lung cancer metastasis (using serum exosomes) [131].

### 6.3. Prognosis and Real-Time Monitoring

An ever-growing number of studies indicate that miRNAs and other ncRNAs promote cancer progression. For instance, miRNA signatures found in tumor tissue, serum and plasma exosomes have aided the prognostication and recurrence prediction of lung cancer [132,133,134], leukemia [154], colon cancer [135], cervical cancer [136] and osteosarcoma [137]. Using a 10-miRNA classifier derived from breast cancer patient tissue biopsy, the recurrence of hormone receptor-positive (HR+) human epidermal growth factor receptor 2 (HER2)-breast cancer can be accurately predicted [138]. In another study using plasma samples from breast cancer patients, dysregulated expression of miRNA-10b and miRNA-373 was found to predict lymph node metastasis [139]. Notably, the overexpression of a single miRNA, miR-210, in tumor biopsy has been shown to be associated with higher risk of recurrence and poorer relapse-free survival of breast cancer patients [140].

Multiple meta-analyses have also demonstrated the prognostic value of other ncRNAs such as lncRNAs, circRNAs and large intergenic noncoding RNA (lincRNA) in cancer patients. For example, expression of a lncRNAs, such as nuclear paraspeckle assembly transcript 1 (NEAT1) [155], myocardial infarction associated transcript (MIAT) [156], noncoding RNA activated by DNA damage (NORAD) [157] and growth arrest-specific 5 transcript (GAS5) [158], as well as expression of circRNA, such as, ciRS-7 [159] and lincRNA-regulator of reprogramming (lincRNA-ROR) [160], have been found to be associated with overall survival of cancer patients.

### 6.4. Predicting Response to Treatment/Treatment Selection/Precision Oncology

In the era of cancer precision medicine, precise characterization of tumors and their surrogate biofluids is necessary to addresses inter-individual variability so as to formulate the most effective treatment for each patient. Many molecular markers have been identified to predict treatment response and also serve as targets for development of novel therapeutics [161]. Among them, miRNAs have been frequently exploited for predicting response to therapies. For example, resistance of tumor cells to gemcitabine treatment and overall survival of pancreatic cancer patients can be predicted by the expression of miR-21 in tumor tissue biopsy [141].

MiRNA profiling was also shown to differentiate colon cancers according to KRAS mutation status, suggesting the existence of mutant KRAS-specific miRNA signatures [142]. Furthermore, miRNA classifiers from tumor- and blood-based liquid biopsy have been identified for treatment response prediction of colorectal cancer (CRC) [51,143,144,145,146] and non-small cell lung carcinoma (NSCLC) [147] patients, respectively. MiRNA profiling also enables treatment response prediction of gliomas (reviewed in [148]), as well as treatment selection for epithelial ovarian cancer [149]. More recently, lncRNAs extracted from liquid biopsies have also been found to be valuable biomarkers for NSCLC diagnosis and prognosis [19].

### 6.5. Minimal Residual Disease

Minimal residual disease (MRD) is defined as a small population of cancer cells that remain in the body after cancer treatment [162]. These cancer cells may remain undetected in routinely used imaging modalities and clinical examinations, leading to tumor relapse or recurrence. Diagnosis of MRD in patients who have had solid tumors poses greater challenge to oncologists due to inaccessibility of the tumor site or diminishing tumor availability over time with chemotherapy (reviewed in [163] for lung cancer). If the MRD can be detected early via longitudinal sampling of minimally invasive liquid biopsies, pre-emptive treatment that targets the MRD can be prescribed to improve progression-free survival.

Currently, MRD testing is mainly used to monitor blood cancers (leukemia, lymphoma and myeloma), where multiparameter flow cytometry (MFC) [164] and quantitative polymerase chain reaction (PCR) [165] are considered to be the standard method of MRD detection. Lately, NGS analysis of tumor-derived fragmentary DNA or circulating tumor DNA (ctDNA) in liquid biopsies longitudinally obtained from patients with acute myeloid leukemia and myelodysplastic syndrome (AML/MDS) has also been shown to enable cancer prognostication [166]. This is consistent with recent data generated by the Memorial Sloan Kettering-Analysis of Circulating cfDNA to Examine Somatic Status (MSK-ACCESS), an NGS assay for detection of very low-frequency somatic alterations in 129 genes in 681 clinical blood samples from 617 patients across 31 distinct solid tumor types [167]. Similarly, other molecular analytes, such as ncRNAs, may also serve as good biomarkers for MRD detection. Although more work is necessary to test this fledgling idea, two groups have demonstrated the utility of circulatory miRNAs in MRD testing of acute lymphoblastic leukemia [150] and chronic myeloid leukemia (CML) [151].

## 7. A Need for Standardization to Enable Precision Medicine

The translation of ncRNA findings from bench-to-bedside has been slow, in part due to differences in how clinical cohort studies were conducted. Variations in study design, selection of study participants, sample size, biospecimen type, isolation procedure, molecular profiling approach and data analytic used appear to be inevitable [168,169,170] (Figure 2). For instance, study designs of various clinical investigations of the same cancer type can vary dramatically, thereby hindering the application of robust meta-analyses to generate valuable insights to disease pathophysiology. Future studies involving the determination of the diagnostic accuracy of ncRNA biomarkers can benefit from the use of several good study design elements, including consecutive enrollment of participants with uniform inclusion and exclusion criteria, blinded testing and interpretation, establishment of pre-specified thresholds, the use of one reference standard for all subjects and the application of relevant statistical analyses [171].

Many clinical oncology studies are also confounded with a myriad of study participant-related variables such as age, race, ethnicity (reviewed in [172]), gender [173,174], stage of cancer, cancer risk factors such as smoking [175], comorbidities as well as concomitant medications. These studies often involved a wide range of biological samples such as blood, serum, plasma, etc. Although liquid biopsy can offer a plethora of advantages, one needs to be mindful of the type of sample used and the limitation associated with it. For example, the abundance of miRNA is known to vary in different biospecimens including solid tissue, blood and other biofluids [176]. Additionally, other critical factors, such as (1) when the sample is collected (e.g., morning versus evening, or pre- versus post-surgery/treatment), (2) type of anti-coagulant used, (3) presence of hemolysis (for biofluids) [177] and/or contaminants [173,177], (4) lack of standardized procedure for sample selection, preparation and processing (whether reported or not) and (5) technical competency of lab personnel should also be carefully dealt with. These variables, if left unchecked by process standardization, could ultimately lead to a data reproducibility crisis and diminish the clinical utility of any promising biomarker detection assays.

Although it is no longer a daunting task to generate big data using the NGS, microarray and RT-qPCR approaches, the lack of data concordance for the same sample among these technology platforms remains a key concern. This is exemplified by the observations derived from human miRnome, where up to 2500 human miRNAs have been discovered by NGS, but only less than a third of them (the so-called “high confidence miRNAs”) can be cross-validated by the other platforms [12]. Clearly, better bioinformatics tools can aid in weeding out false positives that have been incorrectly identified by the NGS approach.

To further enhance the bench-to-bedside translatability of experimental findings, standardization of the conduct of large multi-center prospective clinical studies, as well as pre-analytical preparation of clinical samples prior to storage in established biobanks are needed. The development of specific reporting guidelines (e.g., CONSORT for randomized controlled studies) could also potentially improve data reporting to help others better understand the study design and to assess the validity of findings [178]. Furthermore, as ncRNA biology remains a nascent field that is not well-understood, the sequence and function(s) of many ncRNAs are yet to be elucidated. Ongoing research could focus on building an open access molecular atlas to deepen the understanding on the baseline expression pattern of ncRNAs in various organs, tissues and even distinct cell types, as well as documenting variations in ncRNA expression that are associated with other physiological and pathological conditions, including organ-related injuries or inflammation after surgery [179].

## 8. Leveraging Artificial Intelligence/Machine Learning to Drive Precision Health

The amalgamation of critical disease-associated information from a multitude of molecular analytes (also known as multi-omics) and imaging modalities helps to decipher the complexity of biological networks that drives cancer and other human diseases, but it is frequently a rate-limiting step. Artificial intelligence (AI) or machine learning (ML) algorithms have gradually emerged with the promise that these tools will aid in mining and integrating huge amounts of data generated by multi-omics profiling. Coupled with radiomics and patient-related demographics, clinical and epidemiological factors, one can now build prediction models from a multi-dimensional perspective to enable preventive and precision medicine.

As exemplified by the multi-analyte profiling of single exosomes, the generation of huge amounts of data is becoming a norm and routinely used statistics may no longer be adequate for data management, analysis and interpretation [180]. The use of AI tools to mine useful information from the available molecular and clinical data has gained significant momentum over the past decade. AI/ML not only provides a robust set of tools to combine multiple factors to detect or predict the disease, but it can also be harnessed to assess the extent of contribution from each factor. Such computational approaches have been shown to enable better prediction of CVDs [181].

Given that AI/ML tools are poised to generate wide-ranging impacts in healthcare for patient management, it is of paramount importance to ensure that AI/ML analytics are robust. Many of the challenges in this field are not uniquely associated with ncRNA or liquid biopsy, but more generally encountered in biomedical research. From a technical perspective, it is often tricky to assess the model performance of AI/ML. To bridge the implementation gap of AI/ML in healthcare, various strategies have been devised to counter information leakage problems during model performance assessment. Information leakage can happen during feature selection, where features are selected from both training and testing datasets [182]. Furthermore, information leakage may occur when training and testing datasets are pre-processed together or imputed together [183].

A subset of samples may also be duplicated in the training and testing datasets, and when sampling is performed on the overall dataset, the risk of information leakage is heightened [184]. In addition, time-series data need to be cross-validated differently, where the model should not contain future information prior to assessment [185]. To this end, various schemas have been proposed with the aim to mitigate information leakage in a systematic manner. For example, Richard Simon proposed the use of nested cross-validation [186] for model performance evaluation that has now been widely accepted to yield robust estimation. Poldrack et al. also recommended the use and reporting of multiple measures of model performance [183]. Ultimately, an increase in awareness of these issues [187] and the advocacy to publish source code is slated to further mitigate the erroneous reporting of models.

In the context of biomedical research, many other challenges remain to be addressed when applying AI/ML. As models can only be as good as its training data, we should emphasize generating good wet lab data through better technologies in sample processing and analyte measurements, as well as ensuring high-quality ground-truth labeling of data for training. Furthermore, we need to overcome the shortcomings of current gold standards for disease diagnosis/prognosis. For instance, endoscopic tissue biopsy and post-mortem examination of the brain tissue have been the gold standards for gastric cancer diagnosis [188] and dementia confirmation [189], respectively. While the former is invasive to the patient and difficult to obtain sufficient biospecimens for downstream clinical assays, the latter does not allow for the real-time monitoring of disease progression. Notably, gastroendoscopy has been reported to miss a significant proportion of early-stage gastric cancer when cross-sectional sampling was performed [190], thus highlighting the need to conduct longitudinal sampling via periodic follow-ups after the initial diagnosis to closely monitor at-risk individuals. Given that these standard-of-care diagnostics for complex diseases are often imperfect and inadequate, time-consuming adjudication by a panel of experts may be necessary to mitigate such issues.

Lastly, the lack of prior knowledge of many ncRNAs (sequence and function) have significantly hindered the implementation of AI/ML to assist the adoption of ncRNA-powered in vitro diagnostics (IVDs). As ncRNA-based IVDs represent an uncharted but exciting area of development, their clinical utility as biomarkers is often extrapolated from statistical inferences of reported case control studies. While this approach remains invaluable at the initial phase to generate hypotheses, it is frequently plagued by potential bias in sampling from the case and control arms. Latent variables that are invisible to the study design may further complicate the interpretation of results. Collectively, these issues can be addressed by formulating a more robust study plan and conducting prospective multi-center studies.

## 9. Conclusions

To date, a growing number of studies have demonstrated the clinical utility of tissue and liquid biopsy-based ncRNA biomarkers in the clinical care continuum. As ncRNomics is a relatively nascent field, various challenges and knowledge gaps have been anticipated. By overcoming these limitations, it is envisaged that ncRNAs can ultimately fulfil their potential to become the next standard-of-care clinical biomarkers. Harnessing information from disease-associated ncRNome in biofluids or in tissues will undoubtedly fuel the community adoption of precision population health.

## Figures and Tables

**Figure 1 ijms-23-10390-f001:**
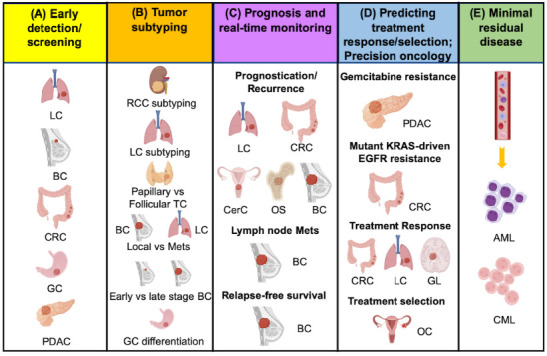
The continuum of cancer care using ncRNA biomarkers [14,15,51,113,114,115,116,117,118,119,120,121,122,123,124,125,126,127,128,129,130,131,132,133,134,135,136,137,138,139,140,141,142,143,144,145,146,147,148,149,150,151]. It includes (**A**) early detection/screening; (**B**) tumor subtyping; (**C**) prognosis and real-time monitoring; (**D**) predicting treatment response/selection; precision oncology; (**E**) minimal residual disease. AML: Acute lymphoblastic leukemia; BC: breast cancer; CRC: colorectal cancer; CerC: cervical cancer; CML: chronic myeloid leukemia; OC: ovarian cancer; GC: gastric cancer; GL: gliomas; LC: lung cancer; NSCLC: non-small cell lung cancer; OS: osteosarcoma; PC: prostate cancer; PDA: pancreatic ductal adenocarcinoma; RCC: renal cell carcinoma; TC: thyroid cancer. This figure is created with BioRender.com.

**Figure 2 ijms-23-10390-f002:**
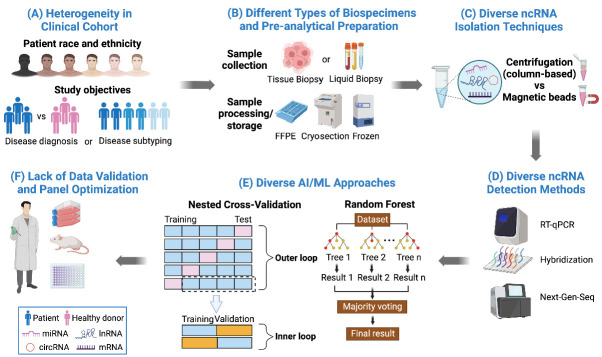
Key challenges to translate ncRNA findings from bench-to-bedside. (**A**) Diverse patient race and ethnicity (White, African American, Asian, American Indian or Alaska Native, Native Hawaiian, etc.) may lead to heterogeneity in clinical cohort based on their genetic background. Various study designs for disease diagnosis or disease subtyping may contribute to data reproducibility issues. (**B**) The expression of ncRNAs is known to vary in different biospecimens including solid tissue, blood and other biofluids. Different sample processing or storage methods may also affect the stability of ncRNA biomarkers. (**C**) A multitude of lab approaches are used for ncRNA isolation in different studies and data accuracy may be affected by RNA yield, composition and purity. (**D**) RT-qPCR, hybridization and NGS represent the three mainstream ncRNA detection techniques, but they differ in ncRNA profiling performance and sensitivity. (**E**) Various AI/ML tools are available to drive big data analytics, but they may yield different results. (**F**) There is also a lack of data validation and panel optimization to ensure robustness of the developed ncRNA assays. FFPE: Formalin-Fixed Paraffin-Embedded; Next-Gen-Seq (NGS): next-generation sequencing; AI: artificial intelligence; ML: machine learning. This figure was created with BioRender.com.

**Table 1 ijms-23-10390-t001:** Sampling methods and biomarker detection in cancer care: advantages and disadvantages. * The use of ncRNA biomarkers in cancer care is largely exploratory in nature due to: (1) many ncRNAs have yet to be identified, (2) functional characterization of known ncRNAs remains incomplete, (3) lack of process standardization to mitigate confounding effect of various factors and (4) isolation and characterization of exosome also remain challenging.

Sampling Methods and Biomarker	Advantages	Disadvantages
Tumor biopsy: Detection of cancer cells by histology	Current gold standard for histological diagnosis and staging of cancer	False positivity associated with formalin-preserved tissues Time-consuming Not cost-effective Inability to assess tumor heterogeneity Requires trained personnel and facilities for surgical and histological procedures Invasive and can lead to surgical complications/pain Sample availability issues as tumor may be inaccessible Less feasible to conduct longitudinal monitoring
Tumor biopsy: Detection of ncRNAs by molecular techniques	Excellent analyte stability (e.g., miRNAs and circRNAs are relatively stable) Possible to assess tumor heterogeneity Possible to amplify biomarker signal for detection Possible to multiplex with other molecular analytes for multicomponent analysis	Exploratory in nature * Lack of standardization False positivity associated with formalin-preserved tissues Invasive and can lead to surgical complications/pain Sample availability issues as tumor may be inaccessible Less cost-effective Less feasible to conduct longitudinal monitoring Requires trained personnel and facilities to perform molecular assays and interpreting the results
Liquid biopsy: Detection of ncRNAs by molecular techniques (including exosome enrichment)	Excellent analyte stability (e.g., miRNAs and circRNAs are relatively stable) Time-saving May potentially be cost-effective Possible to assess tumor heterogeneity Possible to increase biomarker signal-to-noise ratio (through exosome enrichment) Possible to amplify biomarker signal for detection Lower risk of complications/pain Feasible to conduct longitudinal monitoring Small sample volume required Possible to multiplex with other molecular analytes for multicomponent analysis	Exploratory in nature * Lack of standardization Requires trained personnel and facilities to perform molecular assays and interpreting the results

## Data Availability

Not applicable.

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
