# Peer review of "Noncoding RNome as Enabling Biomarkers for Precision Health"

_ijms, 2022, doi:10.3390/ijms231810390_

Round 1

Reviewer 1 Report

This is a well-written review. Cheong et al, bring very relevant issues on the non-coding RNA and biomarker field. 
One minor comment is about the use of the word to classify individuals from different ancestry background. The authors should avoid such kind terms in text.

Author Response

Reviewer 1

Comments and Suggestions for Authors

This is a well-written review. Cheong et al, bring very relevant issues on the non-coding RNA and biomarker field.

One minor comment is about the use of the word to classify individuals from different ancestry background. The authors should avoid such kind terms in text.

Response: We thank the reviewer for his/her compliment and comment. We have revised the figure legend of Figure 2 (“ancestry” is replaced by “background”).

Reviewer 2 Report

In this manuscript Cheong et al provide an excellent review on the role of non-coding RNAs (ncRNAs) as biomarkers. Starting with a general introduction on ncRNAs in general, with a particular emphasis on miRNAs and lncRNAs, the authors discuss the advantages of molecular profiling from biofluids and in particular exosomes profiling, highlighting how exosomes represent ideal vectors for biomarkers.

The authors explain in details the clinical applications of using exosomal ncRNAs in order to diagnose specific diseases, underlining the related challenges to translate ncRNA findings from bench-to-bedside. In addition, the paragraph on the usage of artificial intelligence and machine learning  is particularly informative.

Overall this review is extremely well written and organized with clear tables and figures that effectively summarize the key concepts that the authors intend to convey.

Author Response

Reviewer 2

Comments and Suggestions for Authors

In this manuscript Cheong et al provide an excellent review on the role of non-coding RNAs (ncRNAs) as biomarkers. Starting with a general introduction on ncRNAs in general, with a particular emphasis on miRNAs and lncRNAs, the authors discuss the advantages of molecular profiling from biofluids and in particular exosomes profiling, highlighting how exosomes represent ideal vectors for biomarkers.

The authors explain in details the clinical applications of using exosomal ncRNAs in order to diagnose specific diseases, underlining the related challenges to translate ncRNA findings from bench-to-bedside. In addition, the paragraph on the usage of artificial intelligence and machine learning is particularly informative.

Overall this review is extremely well written and organized with clear tables and figures that effectively summarize the key concepts that the authors intend to convey.

Response: We thank the reviewer for his/her compliment and comment.

Reviewer 3 Report

Review IJMS 1887119 v1

Title: Noncoding RNome as enabling biomarkers for Precision Health

The manuscript is well written and reviews the use of RNome as biomarkers in medicine targeted to specific diseases and people.  The review will be interesting for the readers of IJMS. A few constructive comments are below.

Title: Capitalize “enabling” and “biomarkers”

Throughout, refrain from the use of the first person.

Throughout, the paragraphs are a bit long.  Suggest splitting them up where possible.

Abstract:

Line 21: “place a comma before “as well”

Line 25:  Precision health should be defined, and precision population health should encompass that definition.

Line 34:  “were” rather than “was”

Line 38:  Replace “made” with “synthesized”

Line 50:  “found two decades ago”

Line 54:  “Critical factors” is unclear.  Suggest adding at least one example.  Perhaps molecular stability?

Line 74: “high confidence” is jargon.  Explain.  Clinically validated?

Line 88:  First use of circRNA, thus spell out.

Line 98: comma before “but”

Table 1 heading; Lines 251-255.  The table heading is not descriptive of the table contents.

Table 1:  Some reformatting needed

Line138: CancerSeek needs more information – manufacturer, city, state?

Line 145:  Nice list of critical factors – suggest including those in abstract.

Fig. 1.  The numbers need an explanation – citations?  Also the acronyms such as “LC” for lung cancer should be written out.  The figure should be larger.

Line 309.  Commas: “,like miRNAs,”

Figure 2 needs to be larger.

Author Response

Reviewer 3

Comments and Suggestions for Authors

The manuscript is well written and reviews the use of RNome as biomarkers in medicine targeted to specific diseases and people.  The review will be interesting for the readers of IJMS. A few constructive comments are below.

Response: We thank the reviewer for his/her thorough review of our work and constructive comments. We have addressed the comments as below.

Comment1: Title: Capitalize “enabling” and “biomarkers”

Response: We have updated the title as suggested.

Comment2: Throughout, refrain from the use of the first person.

Response: We have revised the manuscript all throughout to avoid using the first person.

Comment3: Throughout, the paragraphs are a bit long.  Suggest splitting them up where possible.

Response: We have revised the manuscript by splitting up sentences where possible to minimize long paragraphs.

Abstract:

Comment4: Line 21: “place a comma before “as well”

Response: The sentence is updated as suggested.

Comment5: Line 25:  Precision health should be defined, and precision population health should encompass that definition.

Response: Due to the word limit of the abstract section, we have included the definitions of precision health and precision population health in the first paragraph of the Introduction section of our revised manuscript instead.

Comment6: Line 34: “were” rather than “was”

Response: The sentence is updated as suggested.

Comment7: Line 38:  Replace “made” with “synthesized”

Response: The sentence is updated as suggested.

Comment8: Line 50: “found two decades ago”

Response: The sentence is updated as suggested.

Comment9: Line 54: “Critical factors” is unclear.  Suggest adding at least one example.  Perhaps molecular stability?

Response: The above comment refers to the concluding section of the Introduction which states the objectives of the review and hence the ‘critical factors’ were not elaborated. However, to address the comment, we have updated the statement referred in the above comment to include one of the critical factors. The other examples of ‘critical factors’ are elaborated under the specific sections, ncRNAs section in lines 77-79 and liquid biopsy section in lines 146-148.

Comment10: Line 74: “high confidence” is jargon.  Explain.  Clinically validated?

Response:  We have edited the statement accordingly. We removed “high confidence” and rephrased the sentence.

Comment11: Line 88:  First use of circRNA, thus spell out.

Response: circRNA has been spelled out in the original text, line 70 (now line 75).

Comment12: Line 98: comma before “but”

Response: The sentence is updated as suggested.

Comment13: Table 1 heading; Lines 251-255.  The table heading is not descriptive of the table contents.

Response: We have made minor edits in the title of Table 1. We hope it addresses the concern raised in the comment.

Comment14: Table 1:  Some reformatting needed

Response: The manuscript is prepared using the template provided by the journal. With the use of template, we do not see much opportunity to improve the layout and fitting of the table within the manuscript. However, we tried some formatting in the bullet spacing.

Comment15: Line138: CancerSeek needs more information – manufacturer, city, state?

Response: We have cited the work by Cohen et al at this end of this sentence (reference 47). CancerSEEK is the name of the multi-cancer early detection blood test.

Comment16: Line 145:  Nice list of critical factors – suggest including those in abstract.

Response: Thank you for your feedback. There is a word limit for the abstract, and we are unable to accommodate them in the abstract.

Comment17: Fig. 1.  The numbers need an explanation – citations? Also, the acronyms such as “LC” for lung cancer should be written out.  The figure should be larger.

Response: The numbers in the parentheses indicate reference citation. This is added in the figure legend. The acronym “LC: Lung cancer” is already included with other acronyms listed. We believe since the figure is pasted as an image in the word document it appears small. We have tried our best to accommodate all the pictures and text with larger font in the figure. Hope the content is still readable.

Comment18: Line 309.  Commas: “, like miRNAs,”

Response: The sentence is updated as suggested.

Comment19: Figure 2 needs to be larger.

Response: We have tried to revise the figure with increased font size for text.